# FROM SCORES TO PREFERENCES: REDEFINING MOS BENCHMARKING FOR SPEECH QUALITY REWARD MODELING

## ABSTRACT

Assessing the perceptual quality of synthetic speech is crucial for guiding the development and refinement of speech generation models. However, it has traditionally relied on human subjective ratings such as the Mean Opinion Score (MOS), which depend on manual annotations and often suffer from inconsistent rating standards and poor reproducibility. To address these limitations, we introduce MOS-RMBench, a unified benchmark that reformulates diverse MOS datasets into a preference-comparison setting, enabling rigorous evaluation across different datasets. Building on MOS-RMBench, we systematically construct and evaluate three paradigms for reward modeling: scalar reward models, semi-scalar reward models, and generative reward models (GRMs). Our experiments reveal three key findings: (1) scalar models achieve the strongest overall performance, consistently exceeding 74% accuracy; (2) most models perform considerably worse on synthetic speech than on human speech; and (3) all models struggle on pairs with very small MOS differences. To improve performance on these challenging pairs, we propose a MOS-aware GRM that incorporates an MOS-difference-based reward function, enabling the model to adaptively scale rewards according to the difficulty of each sample pair. Experimental results show that the MOS-aware GRM significantly improves fine-grained quality discrimination and narrows the gap with scalar models on the most challenging cases. We hope this work will establish both a benchmark and a methodological framework to foster more rigorous and scalable research in automatic speech quality assessment.

## 1 INTRODUCTION

Assessing the perceptual quality of synthetic speech is crucial for guiding the development and refinement of speech generation models (Valentini-Botinhao & Yamagishi, 2018). The rapid progress of text-to-speech (TTS) and generative audio models has dramatically improved the naturalness and expressiveness of synthetic speech (Wang et al., 2025b; Xu et al., 2025a), but also created new challenges for evaluating speech quality at scale (Lo et al., 2019; Huang et al., 2022).

However, perceptual quality assessment has traditionally relied on human subjective ratings such as the Mean Opinion Score (MOS) (Sector, 1996), which depend on manual annotations and often suffer from inconsistent rating standards and poor reproducibility. MOS-based evaluation asks listeners to rate the quality of speech samples on a fixed scale and averages the scores to produce ground-truth labels. While this approach has been the de facto standard for decades (Sector, 1996), it is expensive to collect, subject to human variability, and difficult to scale for modern systems that generate massive amounts of data (Mittag & Möller, 2019). Furthermore, variations in listener populations and rating procedures lead to inconsistent subjective standards, hindering reproducibility and cross-dataset comparison (Pieper & Voran, 2024).

To address these limitations, we introduce MOS-RMBench (see Figure 1), a unified benchmark that reformulates diverse MOS datasets into a preference-comparison setting, enabling rigorous evaluation across different datasets. By transforming MOS scores into pairwise preference labels, MOS-RMBench eliminates scale inconsistencies and allows models to be trained and evaluated under a consistent comparison framework.

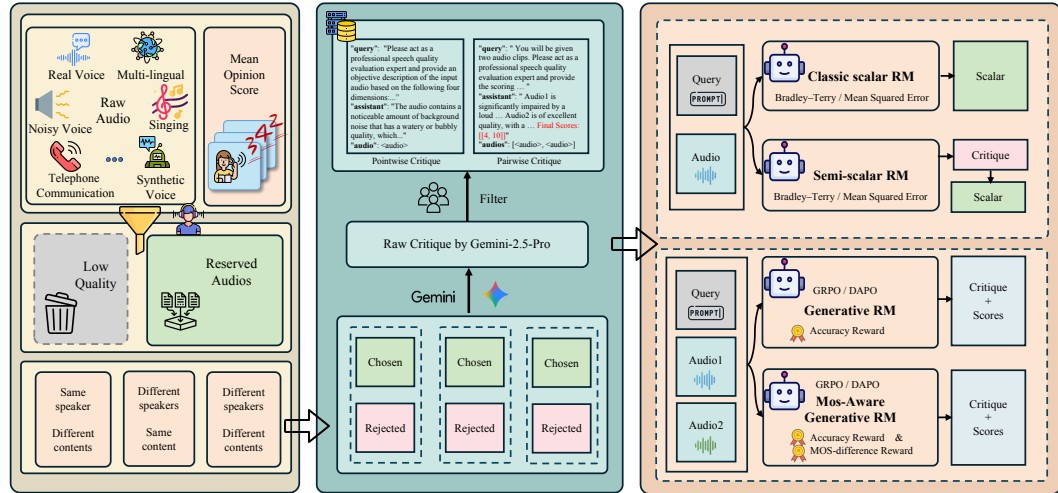

Figure 1: Overview of MOS-RMBench. Data from multiple MOS datasets are filtered and grouped, then converted into pairwise comparisons with natural-language critiques generated by Gemini-2.5-Pro. The resulting dataset supports training and evaluation of reward models in a consistent and reproducible setting.

Building on MOS-RMBench, we systematically construct and evaluate three paradigms for reward modeling: scalar reward models, semi-scalar reward models, and generative reward models (GRMs). First, we find that scalar models achieve the strongest overall performance, consistently exceeding 74% accuracy and reaching around 80% on average across both in-domain and out-of-domain (OOD) datasets, establishing a solid baseline for future work. Second, we observe that most models perform considerably worse on synthetic speech datasets such as SOMOS (Maniati et al., 2022) and VMC'23 (Cooper et al., 2023) compared to human speech, underscoring a persistent domain gap. Finally, although all models struggle on pairs with very small MOS differences, they perform very well when the MOS gap is large, indicating that the key challenge lies in fine-grained quality discrimination.

To improve performance on these challenging pairs, we propose a MOS-aware GRM that augments the standard correctness-based reward with an additional MOS-difference–based reward, enabling the model to adaptively scale rewards according to the difficulty of each sample pair. Experimental results show that the MOS-aware GRM consistently improves performance across all evaluated datasets, with accuracy gains exceeding 3% on samples with particularly similar speech quality.

We hope this work will establish both a benchmark and a methodological framework to foster more rigorous and scalable research in automatic speech quality assessment. Overall, our contributions are threefold:

1. We present MOS-RMBench, the first unified benchmark that reformulates heterogeneous MOS datasets into a consistent preference-comparison framework, enabling rigorous cross-dataset evaluation.
2. We conduct a systematic evaluation of reward modeling paradigms, comparing scalar, semi-scalar, and generative models across diverse speech domains, and providing insights into their relative strengths and weaknesses.
3. We introduce a MOS-aware GRM that adaptively scales pairwise rewards based on MOS differences, achieving measurable improvements in fine-grained speech quality discrimination.

## 2 RELATED WORK

### 2.1 AUTOMATIC SPEECH QUALITY ASSESSMENT

Research on automated speech quality assessment has evolved from early CNN- and RNN-based predictors such as MOSNet and Quality-Net (Lo et al., 2019; Fu et al., 2018) to self-supervised

learning approaches based on wav2vec 2.0 and HuBERT (Baevski et al., 2020; Hsu et al., 2021), and more recently to Audio Language Models (ALMs) (Wang et al., 2025a; Deshmukh et al., 2024; Chen et al., 2025; Zezario et al., 2025). Recent state-of-the-art MOS prediction systems, including UTMOS (Saeki et al., 2022) and LE-SSL-MOS (Qi et al., 2023), have shown strong results on in-domain and OOD tracks of the VoiceMOS challenges. To address the inconsistencies in subjective ratings across datasets, recent work has explored bias-aware training losses (Mittag et al., 2021b) and dataset score alignment frameworks such as AlignNet (Pieper & Voran, 2024). While these methods improve cross-dataset performance, progress remains limited by heterogeneous MOS annotation protocols, motivating the need for a unified benchmark for consistent evaluation.

## 2.2 Advances in Reward Modeling

Recent advances in reward modeling (RM), originally introduced to align model outputs with human preferences (Ouyang et al., 2022), have led to diverse paradigms in text and vision. In the text domain, research has progressed from scalar reward models to generative reward modeling. The Critique-out-Loud (CLoud) framework (Ankner et al., 2024) introduces natural language critiques prior to scalar scoring, bridging generative judgment and reward modeling. More recently, Liu et al. (2025) proposes Self-Principled Critique Tuning (SPCT), enabling GRMs to generate adaptive principles and critiques during inference, achieving state-of-the-art results across RM benchmarks. In the vision domain, RM has been extended to multimodal evaluation, with proprietary systems such as GPT-4V (Achiam et al., 2023) demonstrating strong agreement with human judgments and open-source efforts like LLaVA-Critic (Xiong et al., 2025) unifying pointwise and pairwise scoring. By contrast, research on speech RM remains sparse, with no established frameworks for scalable or fine-grained preference alignment.

## 3 MOS-RMBench-Data

To systematically evaluate reward models for speech quality assessment, we construct MOS-RMBench, a large-scale benchmark centered on Mean Opinion Score (MOS) and preference supervision. MOS-RMBench integrates diverse MOS datasets into a consistent format based on preferences, addresses inconsistencies in scoring standards by using preference pairs rather than absolute MOS scores, and covers diverse speech scenarios. This chapter introduces the benchmark in three aspects: data sources, the annotation strategy for preference data, and overall dataset statistics.

### 3.1 Dataset Source

MOS-RMBench is constructed based on several widely used speech quality datasets. Specifically, we select BVCC (Cooper & Yamagishi, 2021), NISQA (Mittag et al., 2021a), SingMOS (Tang et al., 2024), SOMOS (Maniati et al., 2022), and TMHINT-QI (Chen & Tsao, 2021) as the primary sources for training and in-domain evaluation. To further assess the cross-domain generalization of models, we incorporate the VMC'23 Cooper et al. (2023) dataset as an OOD evaluation benchmark. A brief description of each dataset is provided below:

**BVCC:** The BVCC dataset originates from large-scale listening tests and comprises 7,106 samples, including natural speech and synthetic speech generated by 187 systems spanning diverse TTS and voice conversion (VC) methods. The data sources include Blizzard Challenge, Voice Conversion Challenge, and publicly available samples from ESPnet-TTS (Hayashi et al., 2020).

**NISQA:** NISQA is designed for speech quality assessment in communication networks, covering both simulated distortions and real call recordings. The training and validation sets comprise 11,020 and 2,700 samples, respectively, and consist of simulated and live subsets. The test set contains four subsets, with a total of 952 samples.

**SingMOS:** SingMOS is an open-source, high-quality singing voice MOS dataset in Chinese and Japanese, containing 3,421 segments from both natural singing and 33 systems. It covers various synthesis techniques including singing voice synthesis (SVS), singing voice conversion (SVC), and vocoder-based re-synthesis.

**SOMOS:** SOMOS is a large-scale MOS dataset focusing on neural TTS. It contains 20,100 samples synthesized from LJ Speech (Ito & Johnson, 2017) by 200 neural TTS systems and natural speech,

all generated using the same LPCNet vocoder (Valin & Skoglund, 2019) to isolate acoustic-model differences. MOS-RMBench adopts the SOMOS-clean subset to ensure annotation consistency and reliability.

**TMHINT-QI:** TMHINT-QI is a MOS dataset focused on Mandarin speech, mainly for evaluating speech enhancement (SE) systems. It contains 24,408 samples generated by adding four types of noise (babble, street, pink, white) at four SNR levels (-2, 0, 2, 5 dB) to clean speech, then processed by five SE systems.

**VMC'23:** The VMC'23 dataset originates from The VoiceMOS Challenge 2023 (Cooper et al., 2023). It includes three tracks: (1) French TTS, based on Blizzard Challenge 2023 (Perrotin et al., 2023) listening tests, with 1,460 samples; (2) singing voice conversion, based on the Voice Conversion Challenge 2023 (Huang et al., 2023), containing 4,040 samples; (3) noisy and enhanced speech, based on TMHINT-QI(S) (Zezario et al., 2024), consisting of 1,960 samples.

### 3.2 PREFERENCE ANNOTATION

Given the differences in MOS scoring standards across existing datasets, we adopt a preference-based annotation strategy that transforms MOS scores into a unified supervision signal to achieve consistent evaluation. The core idea is to replace absolute MOS scores with relative preferences between paired samples, because absolute scores are susceptible to dataset-specific biases and inconsistent scoring scales. By formulating the task as preference comparison, MOS-RMBench provides a consistent signal across datasets with diverse sources and evaluation standards, enabling fair and robust benchmarking of reward models for speech quality assessment across domains and scenarios.

We first convert all samples into a unified 16 kHz WAV format to ensure consistency in subsequent processing. Following this, we filter out samples with unreliable annotations, defined as those containing incomplete or misformatted metadata such as speaker, content, or system identifiers, which are essential for constructing pairwise preferences. We then partition the remaining data from each source dataset into three categories: (i) samples that share the same speech content but differ in speaker, synthesis system, or speech processing conditions, for example a TTS system or noise perturbation; (ii) samples that share the same speaker or are generated by the same system but differ in speech content; and (iii) samples that differ in both content and speaker or system. To ensure that comparisons are meaningful and consistent, we construct preference pairs within each dataset based on the intrinsic relationships between samples. Specifically, for the first two categories, we group samples by shared content or by shared speaker or system, respectively, then construct preference pairs within each group and ensure that the two samples in each pair have different MOS scores. The sample with the higher MOS is labeled "chosen", and the one with the lower score is labeled "rejected". For the third category, since explicit grouping is not feasible, we construct preference pairs directly within the dataset and apply the same "chosen-rejected" labeling rule. Finally, we ensure that the number of constructed preference pairs is balanced across datasets so that subsequent evaluations are not biased toward any particular source.

This annotation strategy yields two primary benefits. First, it ensures that pairwise comparisons reflect meaningful contrasts—either differences in system quality under conditions where the content is matched, or differences in content under conditions where the system is matched. Second, by converting absolute MOS values into relative preferences, the strategy reduces sensitivity to dataset-specific scoring standards and annotation biases, which facilitates the integration of heterogeneous MOS sources into a single benchmarking framework. We additionally perform a human verification of the annotated results, as detailed in the Appendix C.1.

### 3.3 DATASET STATISTICS

**Overview.** MOS-RMBench constructs a large-scale and balanced preference dataset comprising 55,333 training samples, 9,905 development samples, and 6,240 in-domain test samples from five source datasets, together with 3,000 OOD test samples from the VMC'23 challenge for assessing cross-domain generalization, as summarized in Table 1. The benchmark covers a broad spectrum of speech scenarios, including natural and synthetic speech, singing voice, VC, SVC, SE, and speech with real or simulated noise and distortions, spanning five languages: English, Chinese (including Mainland Mandarin and Taiwanese Mandarin), Japanese, French, and German. This diversity in

Table 1: Statistics of MOS-RMBench across datasets.

| Dataset | Train | Dev | Test | Scenario | Language |
|---|---|---|---|---|---|
| BVCC | 9,948 | 2,132 | 1,000 | natural speech, TTS, VC | English |
| NISQA | 11,571 | 2,796 | 2,240 | natural & distorted speech | English, German |
| SingMOS | 10,000 | 2,720 | 1,000 | natural speech, SVS, SVC | Chinese, Japanese |
| SOMOS | 13,814 | 2,257 | 1,000 | natural speech, TTS | English |
| TMHINT-QI | 10,000 | – | 1,000 | natural & noisy speech, SE | Chinese |
| VMC'23 | – | – | 3,000 | natural & noisy speech, TTS, SVS, SVC, SE | French, English, Chinese |
| Overall | 55,333 | 9,905 | 9,240 | – | – |

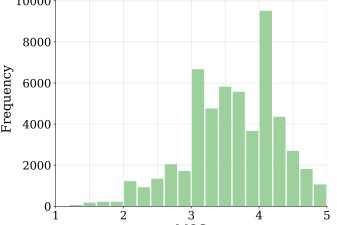 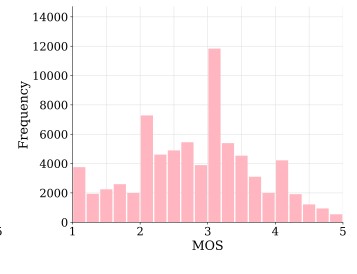 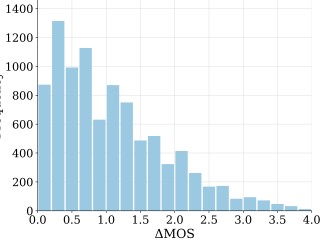

(a) Distribution of chosen samples    (b) Distribution of rejected samples    (c) Distribution of $\Delta MOS$

Figure 2: Distribution of MOS scores in MOS-RMBench. Figures (a) and (b) show the MOS distributions of chosen and rejected samples. Figure (c) presents the distribution of $\Delta MOS$ in the test set.

acoustic conditions and linguistic coverage establishes a rigorous foundation for the development and evaluation of reward models, enabling them to achieve more robust perceptual discrimination and to generalize effectively across domains and languages.

**MOS Distribution.** Figure 2 illustrates the distribution of MOS scores within MOS-RMBench. Overall, the MOS ratings of the chosen samples are predominantly concentrated in the range of 3 and above, whereas the rejected samples are largely distributed below 3.5. This clear separation between the two categories reflects the internal consistency and reliability of the preference annotations. For the test set, the MOS difference ($\Delta MOS$) between each paired sample is mostly within 1.5 points, indicating that a substantial portion of the speech pairs exhibit very similar perceptual quality, which makes the benchmark particularly challenging for speech quality assessment.

## 4 MOS-RMBench-Eval

To comprehensively evaluate the capability of reward models in speech quality assessment, we design MOS-RMBench-Eval. Centered on a unified preference-based setting, the benchmark encompasses diverse reward modeling paradigms, thereby enabling systematic and fair comparison across models. This section introduces the evaluation task and the reward models under evaluation.

### 4.1 EVALUATION TASK

**Task Description.** The evaluation is formulated as a binary preference comparison task: given a pair of speech samples, the model must either assign quality scores to both or directly decide which sample exhibits higher perceptual quality. The model's performance on this task reflects its ability to discriminate differences in speech quality. To reduce potential position bias, the presentation order of the two audio clips in each pair is randomly swapped during evaluation. A decision is counted as correct only if the model assigns a strictly higher score to the sample annotated as chosen than to the one annotated as rejected.

**Evaluation Metrics.** Model performance is reported in terms of accuracy (Acc). For each dataset, we calculate the proportion of evaluation pairs in which the model's scores correctly reflect the

annotated preference (i.e., the chosen sample receives the higher score). The overall accuracy is computed as the proportion of correctly judged pairs across all samples from all datasets, which provides a comprehensive measure of model performance across the diverse speech scenarios covered by MOS-RMBench-Eval.

## 4.2 EVALUATED MODELS

We implement and evaluate a range of reward modeling paradigms based on Qwen2-Audio-7B (Chu et al., 2024), covering scalar, semi-scalar, and GRMs, as well as UTMOS used for MOS prediction and the LLM-as-a-judge (Zheng et al., 2023) paradigm. The following provides a description of the models evaluated under each paradigm, with detailed configurations provided in the Appendix B.

**Classic scalar reward models.** These models output a single quality score for each audio sample. We adopt two training objectives: the Bradley–Terry(BT) (Bradley & Terry, 1952) loss, which maximizes the likelihood that the sample annotated as chosen receives a higher predicted score, and the mean squared error (MSE) loss, which minimizes the squared difference between the predicted score and the corresponding MOS value.

**Semi-scalar reward models.** These models extend the scalar paradigm by incorporating natural language descriptions of audio quality. For each audio sample, Gemini-2.5-Pro (Comanici et al., 2025) provides an overall quality description along four dimensions: noise, distortion, continuity, and naturalness. The model is first trained to generate these descriptions, after which its outputs are passed through a reward head to produce scalar scores. Similar to classic scalar models, both BT and MSE objectives are used for training.

**Generative reward models.** GRMs take two audio samples as input simultaneously. As in the semi-scalar setting, the model first learns to produce descriptive quality assessments. It is then further refined through supervised fine-tuning (SFT) on paired audio data annotated by Gemini-2.5-Pro, where each pair is compared across the same four dimensions and provided with overall quality scores. The annotation is validated to ensure that the chosen sample receives the higher score. GRMs are further optimized using reinforcement learning methods, including Group Relative Policy Optimization (GRPO) (Shao et al., 2024) and Decoupled Clip and Dynamic sAmpling Policy Optimization (DAPO) (Yu et al., 2025). In both cases, training employs a rule-based reward design, referred to as the Accuracy reward, which is determined solely by whether the model correctly judges the quality of a speech sample pair. Using $S(c)$ and $S(r)$ to denote the scores assigned to the chosen and rejected samples, respectively, the Accuracy reward is defined as follows:

$$\text{Accuracy reward} = \begin{cases} 1 & if \ \ S(c) > S(r), \\ -1 & otherwise \end{cases} \tag{1}$$

**LLM-as-a-judge.** These models directly compare two audio samples and output a preference judgment without producing explicit scalar scores. We evaluate this paradigm using Gemini-2.5-Pro and Qwen2.5-Omni-7B (Xu et al., 2025b).

## 5 EXPERIMENT

This section presents an empirical study on MOS-RMBench to investigate the performance of different paradigms. We first present the overall evaluation results, then perform an error analysis across samples with varying MOS gaps, and finally explore a MOS-aware GRM designed to address the limitations revealed by this analysis.

## 5.1 EVALUATION RESULTS

Table 2 presents the main evaluation results, summarizing the performance of all models across individual datasets as well as overall.

**Comparison of different modeling paradigms.** As shown in the evaluation results, the Classic scalar models achieve the highest overall accuracy (80.04% with BT loss), followed by the Cloud semi-scalar models (78.82% with BT loss), while the GRMs attain slightly lower overall performance. These findings indicate that, under the current evaluation setup, the Classic scalar paradigm

Table 2: Evaluation results of models from different paradigms on MOS-RMBench. The best results are shown in **bold** and the second best is with underline. Classic + BT/MSE Loss denote scalar reward models trained with BT or MSE loss; Cloud + BT/MSE Loss denote semi-scalar reward models with the same losses; GRM + SFT/GRPO/DAPO denote generative reward models trained without RL, with GRPO, or with DAPO.

| Model | BVCC | NISQA | SingMOS | SOMOS | TMHINT-QI | VMC'23 | Overall |
|-------|------|-------|---------|-------|-----------|--------|---------|
| *MOS prediction model* | | | | | | | |
| UTMOS | **86.70** | 73.17 | 63.80 | 65.00 | 68.10 | 60.83 | 68.18 |
| *LLM-as-a-judge* | | | | | | | |
| Gemini-2.5-Pro | 63.90 | 81.96 | 58.50 | 64.10 | 73.20 | 65.57 | 69.26 |
| Qwen2.5-Omni-7B | 54.00 | 63.97 | 57.70 | 56.30 | 69.90 | 58.43 | 60.23 |
| *Scalar reward models* | | | | | | | |
| Classic + BT Loss | 85.70 | **83.93** | 74.80 | **76.80** | **81.10** | **77.73** | **80.04** |
| Classic + MSE Loss | 82.80 | 79.33 | 69.80 | 74.30 | 79.40 | 72.23 | 75.83 |
| *Semi-scalar reward models* | | | | | | | |
| CLoud + BT Loss | 85.50 | 81.07 | **78.20** | 75.30 | 80.60 | 75.70 | 78.82 |
| CLoud + MSE Loss | 84.50 | 77.81 | 76.10 | 75.20 | 80.50 | 73.67 | 77.01 |
| *Generative reward models* | | | | | | | |
| GRM + SFT | 80.60 | 79.60 | 75.10 | 70.00 | 77.90 | 69.23 | 74.63 |
| GRM + GRPO | 82.50 | 80.31 | 76.40 | 74.60 | 78.10 | 73.13 | 76.94 |
| GRM + DAPO | 82.60 | 82.05 | 77.50 | 74.40 | 79.30 | 73.13 | 77.60 |

remains highly effective in distinguishing speech quality, despite the additional modeling flexibility offered by the semi-scalar and generative paradigms. Notably, while UTMOS performs strongly on BVCC, it exhibits a marked drop in accuracy on the other datasets, suggesting that MOS prediction models may generalize poorly across diverse datasets. Furthermore, Gemini-2.5-Pro and Qwen2.5-Omni-7B both achieve accuracies below 70% on most datasets, underscoring the considerable challenge that MOS-RMBench presents to current audio large language models. We also conducted a study on preference reversal across different reward modeling paradigms, with detailed results provided in the Appendix C.2.

**Comparison of training objectives for scalar-based reward models.** Within scalar-based paradigms, we examine the effect of different training objectives: BT loss and MSE loss. For the Classic scalar models, BT loss consistently outperforms MSE loss, achieving over a 4% gain in overall accuracy; a similar trend is observed for the Cloud semi-scalar models. The advantage of BT loss over MSE loss is even more pronounced on the OOD VMC'23 dataset. These findings indicate that, compared with direct regression to MOS scores, optimizing the relative ordering of samples with BT loss is generally more effective. The advantage likely stems from BT loss explicitly encouraging the model to capture relative quality differences between samples, whereas MSE is more sensitive to variations in absolute MOS scores across datasets, affecting cross-domain robustness.

**Comparison of training strategies for GRMs.** For GRMs, we evaluate two reinforcement learning strategies: DAPO and GRPO. The two methods attain comparable overall accuracies, with 77.60% for DAPO and 76.94% for GRPO. Although the differences among strategies are modest, DAPO appears to have a slight but consistent advantage in learning from paired audio comparisons on average. More importantly, both strategies yield a clear and consistent performance gain compared with SFT alone, further highlighting the benefit and necessity of reinforcement learning for GRMs. Furthermore, all GRM variants demonstrate competitive performance across both in-domain and OOD datasets, particularly on NISQA and SingMOS, despite their overall accuracy remaining slightly lower than that of the Classic scalar models.

## 5.2 ERROR ANALYSIS

**What constrains model performance?** We observe that all models are particularly prone to errors on sample pairs with small MOS differences. To quantify this phenomenon, we conduct a percentile-

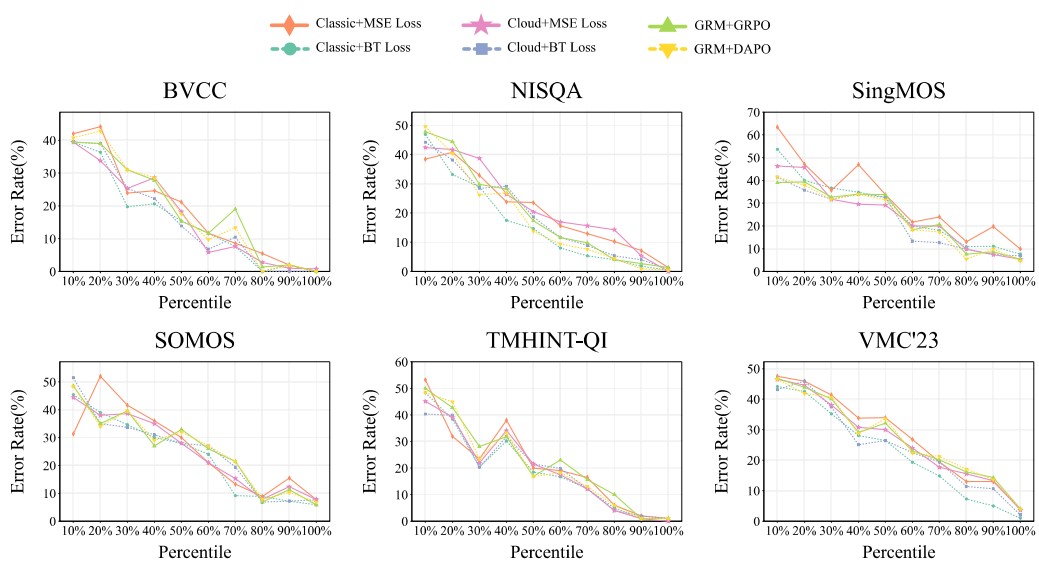

Figure 3: Percentile-based error analysis across datasets: error rates are highest for pairs with small MOS differences and decline markedly as the MOS gap widens.

based error analysis: within each dataset, pairs are first sorted by their MOS difference in ascending order and then divided into percentile bins of equal size. We then compute, for each bin, the proportion of pairs that each reward model ranks incorrectly. Figure 3 presents the statistical results. Across all models, error rates are consistently high in the lowest MOS difference percentiles, even the top-performing Classic scalar model exhibits error rates of 40% or higher. As the MOS difference increases, however, the error rates for all models decrease markedly. These findings indicate that fine-grained discrimination of speech quality remains a critical bottleneck for current reward models, highlighting the need for methods capable of capturing subtle speech quality differences.

## 5.3 MOS-AWARE GRM

**Design of the MOS-aware reward function.** GRMs offer a flexible and interpretable framework for reward modeling, allowing the reward signal to be decomposed and extended. During the original GRM reinforcement learning process, the Accuracy reward considers only whether the model correctly ranks the quality of an audio pair, treating all pairs as equally informative. This uniform treatment overlooks the inherent difficulty of pairs with small MOS differences, which may constrain the model's ability to learn fine-grained quality distinctions. To address this limitation, we introduce the MOS-difference reward, which explicitly incorporates the MOS gap of each audio pair into the reward function:

$$\text{MOS-difference reward} = \begin{cases} 0.5 * (1.0 + cos(\Delta MOS * \pi)) & if \ \ S(c) > S(r), \\ 0.5 * (-1.0 + cos(\Delta MOS * \pi)) & otherwise \end{cases} \quad (2)$$

Specifically, the MOS gap refers to the difference between the original MOS scores of the paired samples in the dataset. $\Delta MOS$ is obtained by normalizing this gap by the 90th percentile of MOS differences in the dataset and clamping the resulting value to the [0,1] range to mitigate the influence of extreme outliers. Based on the Accuracy reward and the MOS-difference reward, the final MOS-aware reward is defined as follows:

$$\text{MOS-aware reward} = \text{Accuracy reward} + \text{MOS-difference reward}. \quad (3)$$

This formulation offers two key advantages. First, it enables adaptive scaling of the reward based on the relative difficulty of each sample pair: for pairs with small MOS differences, correct predictions are assigned a relatively larger reward while incorrect predictions incur a relatively smaller penalty;

Table 3: Evaluation results of MOS-aware GRMs with GRPO and DAPO on MOS-RMBench. Numbers in parentheses denote absolute improvements over baseline GRM.

| Models | BVCC | NISQA | SingMOS | SOMOS | TMHINT-QI | VMC'23 | Overall |
|---|---|---|---|---|---|---|---|
| MOS-aware GRM+GRPO | 83.10 (+0.60) | 81.47 (+1.16) | 76.50 (+0.10) | 75.80 (+1.20) | 80.40 (+2.30) | 74.13 (+1.00) | 78.00 (+1.06) |
| MOS-aware GRM+DAPO | 83.40 (+0.80) | 82.14 (+0.09) | 78.40 (+0.90) | 75.10 (+0.70) | 79.90 (+0.60) | 73.57 (+0.44) | 78.08 (+0.48) |

for pairs with large MOS differences, the reward and penalty are modulated accordingly. Second, the cosine-based shaping ensures smooth transitions at both ends of the scale. By adjusting the reward according to the implied difficulty, the MOS-aware design provides a more informative learning signal, encouraging the model to attend to subtle perceptual distinctions while still penalizing clearly incorrect predictions.

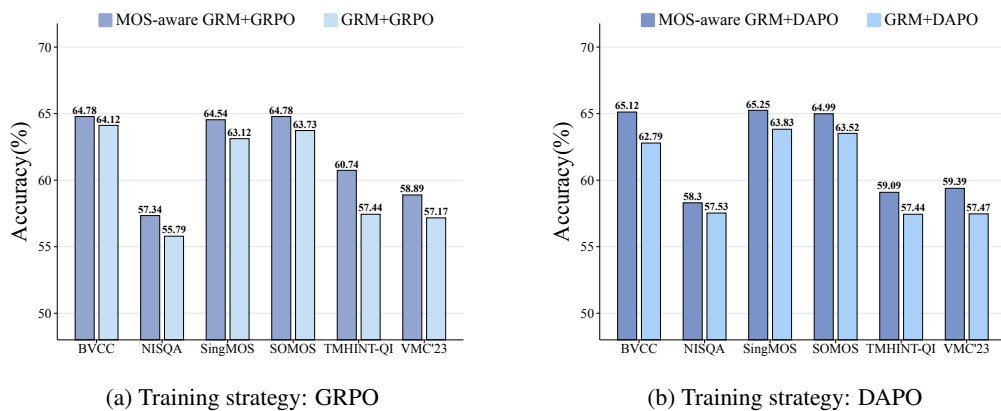

(a) Training strategy: GRPO  (b) Training strategy: DAPO

Figure 4: Performance comparison of MOS-aware GRMs and standard GRMs trained with different reinforcement methods on samples with MOS difference $\leq 0.5$.

**Impact of MOS-aware reward on GRM performance.** To evaluate the effectiveness of the proposed MOS-aware reward, we incorporate it into the GRM training process under both GRPO and DAPO strategies, while keeping all other training configurations unchanged. This design enables a direct comparison with baseline GRM models that rely solely on the Accuracy reward. Table 3 summarizes the overall results, showing that MOS-aware GRMs achieve consistent improvements over their baseline counterparts across all evaluation scenarios.

To further examine performance on perceptually challenging pairs, we evaluate models on sample pairs with a MOS difference threshold of 0.5. Figure 4 presents the corresponding results. Across all datasets, MOS-aware GRMs consistently outperform the baselines on these fine-grained comparisons. For illustration, with the GRPO strategy, accuracy increases from 57.44% to 60.74% on THMINT-QI and from 57.17% to 58.89% on the OOD VMC'23 dataset; under the DAPO strategy accuracy improves from 62.79% to 65.12% on BVCC and from 57.47% to 59.39% on VMC'23. These results demonstrate that incorporating MOS difference information into the reward function produces stable and statistically significant gains. The MOS-aware design provides a more informative, difficulty-sensitive learning signal that strengthens GRM's capacity to capture fine-grained perceptual quality differences beyond what the conventional Accuracy reward achieves. A detailed evaluation of the MOS-aware reward's effectiveness in more challenging scenarios is provided in the Appendix C.3.

## 6 CONCLUSION, LIMITATIONS AND FUTURE WORK

In this work, we introduced MOS-RMBench, the first unified benchmark for speech quality reward modeling that reformulates six major MOS datasets into a preference-based evaluation framework. Our experiments show that scalar reward models achieve the strongest overall accuracy (80%), semi-scalar models perform comparably, and GRMs achieve slightly lower accuracy but offer greater

interpretability. Error analysis reveals that most misrankings occur on pairs with very small MOS differences, highlighting fine-grained quality discrimination as a key challenge.

Despite its coverage, MOS-RMBench is limited to a small set of languages and focuses primarily on perceptual quality, leaving prosody, emotion, and style unmodeled. Reformulating MOS scores into pairwise preferences also changes the original score distribution, which may affect how models learn absolute quality levels.

Future work includes expanding the benchmark to more languages and acoustic conditions, incorporating multi-dimensional quality dimensions, and developing reward models that better capture subtle perceptual differences. We hope MOS-RMBench will serve as a reproducible foundation for advancing speech quality reward modeling and preference alignment in next-generation speech generation systems.

## ETHICS STATEMENT

This study strictly adheres to academic research ethics. All models used are employed in accordance with their respective licenses, and all datasets are publicly available and used in compliance with their terms. The research does not involve human subjects, private or sensitive personal data, or proprietary information, and no experiments pose potential harm to individuals, communities, or the environment.

## REPRODUCIBILITY STATEMENT

We take reproducibility seriously and provide sufficient information to enable others to replicate the results reported in this paper. All datasets used are publicly available, and detailed statistics, data splits, and preprocessing steps are described in the paper. Training configurations, hyperparameters, and evaluation protocols for all models are fully documented, along with the prompts used for data annotation and model evaluation. In addition, we provide the source code and relevant data as supplementary materials to facilitate exact reproduction of our experiments. We will release the code and datasets upon acceptance.

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

## A  USAGE OF LARGE LANGUAGE MODELS

In this study, large language models were used only for polishing parts of the manuscript's text to improve fluency and readability. They did not participate in the research design, the development or execution of the methodology, the collection or analysis of data, or the creation and validation of the core scientific content. All core research content and findings are entirely the work and responsibility of the authors.

## B  EXPERIMENTAL DETAILS

### B.1  MODEL CONFIGURATIONS AND HYPERPARAMETERS

All models are trained in a fully parameterized manner.

**Classic scalar reward models:** Both BT-loss and MSE-loss variants are trained for one epoch with a batch size of 32, using the AdamW optimizer (initial learning rate 5e-6, weight decay 5e-6).

**Semi-scalar reward models:** Both BT-loss and MSE-loss variants are trained for one epoch with a batch size of 32, using the AdamW optimizer (initial learning rate 1e-6, weight decay 1e-6). The total loss is a weighted sum of the reward loss and the LM loss, where the LM loss is assigned a weight of 1.25.

**Generative reward models:** Both GRPO and DAPO are trained for one epoch with a batch size of 64 using the AdamW optimizer (learning rate 1e-6), generating 4 completions per prompt with temperature 1.0, and employing ZeRO-2 optimization.

For inference, the temperature was set to 0.0 for all models. Training and inference are conducted on eight H20 GPUs for all models.

### B.2  PROMPT TEMPLATES

This section details the prompt templates used both for data annotation and for model inference during evaluation.

**Prompts for Data Annotation.** To construct the training sets for the reward models, we used Gemini-2.5-Pro as the annotator and designed two kinds of prompts. For single-sample critic annotation, as illustrated in Figure 5, Gemini-2.5-Pro is guided to listen to a single speech sample and provide a detailed critique of its perceptual quality across multiple dimensions. For paired-sample critic annotation, as shown in Figure 6, Gemini-2.5-Pro compares two speech samples, offers a detailed assessment of each along key perceptual dimensions, and then assigns overall quality scores from 1 to 10 to both samples. This paired-comparison protocol yields both textual rationales and numerical ratings that serve as the foundation for preference-based generative reward modeling.

**Prompts for Model Inference.** During evaluation, different reward-modeling paradigms adopt distinct prompt formats. Figure 7 presents the prompt structure for both the Classic scalar reward models and the semi-scalar reward models: the Classic scalar models assess a single speech sample and output a scalar score representing its overall perceptual quality, whereas the semi-scalar models produce a detailed critique of the sample before deriving a scalar quality score. Figure 8 shows the prompt for the GRMs, which requires the model to compare two speech samples across four perceptual dimensions and then assign an overall quality score from 1 to 10 to each sample. Finally, Figure 9 illustrates the prompt for the LLM-as-a-judge paradigm, where the model is asked to decide

which of two input speech samples has higher overall perceptual quality and to output the identifier of the higher-quality sample.

## C  ADDITIONAL EXPERIMENTAL RESULTS

### C.1  HUMAN VERIFICATION

**Validation of the benchmark.** To assess the reliability of the benchmark's MOS-derived labels, we conducted a human validation study. Four human evaluators annotated "chosen/rejected" labels for 100 randomly sampled preference pairs. The annotations yielded an 82% agreement with the benchmark's MOS-derived labels. This level of agreement indicates that the benchmark closely reflects human perceptual preferences.

**Validation of LLM annotation reliability.** We evaluated the consistency between natural-language critiques generated by Gemini-2.5-Pro and human perceptual judgments. Four human evaluators assessed whether the model's critiques aligned with their own judgments on 100 randomly sampled items. We observed that 70% of the critiques were consistent with human perception. Unlike binary preference judgments, natural-language critiques require describing fine-grained perceptual attributes (e.g., noise, distortion, continuity, naturalness), where subjective variability is inevitable. Under these conditions, a 70% alignment rate is considered reasonable and expected.

### C.2  STUDY OF PREFERENCE REVERSAL

Preference reversal primarily refers to the presence of preference cycles, where a model's pairwise judgments violate transitivity (e.g., A>B, B>C, but C>A). Scalar and semi-scalar reward models assign deterministic scalar scores independently for each sample (with temperature = 0), which inherently prevents the formation of cycles. In contrast, GRMs score each pair directly, making them susceptible to exhibiting such cycles.

To directly assess the occurrence of preference cycles in GRMs, we conducted an explicit triplet-based cycle detection study. For each dataset, we followed the original grouping rules and sampled triplets (A,B,C) with distinct MOS scores such that MOS(A)>MOS(B)>MOS(C), yielding three preference pairs: (A,B), (B,C), and (A,C). Across all datasets, we constructed a total of 565 triplets (1,695 preference pairs), including 65 from SOMOS and 100 from each of the remaining datasets. We then evaluated GRMs trained with SFT, GRPO, and DAPO to quantify both the pairwise accuracy and the preference-cycle rate, as reported in Table 4 and Table 5.

Table 4: Pairwise accuracy (%) of GRMs trained with SFT, GRPO, and DAPO on the preference-cycle evaluation set constructed from sampled triplets.

| Model | BVCC | NISQA | SingMOS | SOMOS | TMHINT-QI | VMC'23 | Overall |
|---|---|---|---|---|---|---|---|
| GRM+SFT | 82.67 | 85.00 | 73.67 | 64.62 | 80.00 | 66.33 | 76.05 |
| GRM+GRPO | 84.67 | 85.33 | 74.67 | 72.31 | 77.33 | 71.33 | 77.94 |
| GRM+DAPO | 84.00 | 87.00 | 75.67 | 70.77 | 81.00 | 72.33 | 78.94 |

Table 5: Preference-cycle rate (%) of GRMs trained with SFT, GRPO, and DAPO on the preference-cycle evaluation set constructed from sampled triplets.

| Model | BVCC | NISQA | SingMOS | SOMOS | TMHINT-QI | VMC'23 | Overall |
|---|---|---|---|---|---|---|---|
| GRM+SFT | 2.00 | 2.00 | 2.00 | 1.54 | 0.00 | 9.00 | 2.83 |
| GRM+GRPO | 0.00 | 0.00 | 0.00 | 0.00 | 0.00 | 1.00 | 0.18 |
| GRM+DAPO | 0.00 | 0.00 | 0.00 | 0.00 | 1.00 | 0.00 | 0.18 |

The results indicate that the SFT-trained GRM exhibits a small yet non-negligible preference-cycle rate (2.83% overall), accompanied by lower pairwise accuracy. In contrast, the RL-trained GRMs

(GRPO and DAPO) achieve substantially higher accuracy and reduce preference cycles to almost zero (0.18% overall for both methods), with only a single cycle observed across all datasets. These findings demonstrate that RL-based training not only enhances alignment with human preferences but also effectively suppresses preference reversals.

## C.3 Effectiveness of the MOS-aware Reward in Challenging Scenarios

To further evaluate the performance of MOS-aware GRMs on more challenging samples, we specifically tested pairs with extremely small MOS gaps ($\Delta MOS < 0.2$). Both GRPO-trained and DAPO-trained GRMs were evaluated, and the MOS-aware GRMs were compared against the corresponding vanilla GRMs. The evaluation results are presented in Table 6and Table 7. Across both training algorithms, the MOS-aware GRMs consistently achieve improved or comparable accuracy on pairs with extremely small MOS differences.

Table 6: Evaluation results of the MOS-aware GRM with GRPO on the subset with $\Delta MOS < 0.2$. The better results in each column are shown in **bold**.

| Model | BVCC | NISQA | SingMOS | SOMOS | TMHINT-QI | VMC'23 | Overall |
|---|---|---|---|---|---|---|---|
| GRM+GRPO | 56.79 | 52.85 | 60.98 | **58.19** | 43.24 | 53.20 | 54.41 |
| MOS-aware GRM+GRPO | **58.02** | **54.40** | **63.41** | 57.63 | **48.65** | **53.78** | **55.33** |

Table 7: Evaluation results of the MOS-aware GRM with DAPO on the subset with $\Delta MOS < 0.2$. The better results in each column are shown in **bold**.

| Model | BVCC | NISQA | SingMOS | SOMOS | TMHINT-QI | VMC'23 | Overall |
|---|---|---|---|---|---|---|---|
| GRM+DAPO | 58.02 | 54.40 | 63.41 | 57.06 | 48.65 | 52.62 | 54.75 |
| MOS-aware GRM+DAPO | 58.02 | **55.44** | 63.41 | **59.32** | 48.65 | **53.49** | **55.78** |

**GRPO setting:** Compared with the vanilla GRM, the MOS-aware GRM increases overall accuracy from 54.41% to 55.33%, with gains observed on BVCC, NISQA, SingMOS, TMHINT-QI, and VMC'23. A slight decrease is noted on SOMOS, but the gap is minor.

**DAPO setting:** The MOS-aware GRM improves overall accuracy from 54.75% to 55.78%, showing equal or better performance across all datasets.

These results demonstrate that incorporating MOS-gap information into the reward function yields measurable benefits, particularly in the fine-grained regime where perceptual differences are extremely small and the task is most challenging.

810
811
812
813
814
815
816
817
818
819
820
821
822
823
824
825
826
827
828
829
830
831
832
833
834
835
836
837
838
839
840
841
842
843
844
845
846
847
848
849
850
851
852
853
854
855
856
857
858
859
860
861
862
863

**Prompt for Gemini-2.5-Pro to annotate a single-audio critic**

**System:**
You are a helpful assistant.

**Prompt:**
## **[Instruction starts]**

You will be given an audio clip.
Please act as a professional speech quality evaluation expert and provide objective
description for the audio clip based on the following four dimensions:
1. Noise: Whether there is background noise, and whether it interferes with understanding.
2. Distortion: Whether there are compression artifacts, electrical noise, or other distortions.
3. Naturalness: Whether the speech sounds natural and resembles real human speech.
4. Continuity: Whether the speech is fluent and continuous, or if there are any dropouts or
interruptions.

## **[Instruction ends]**

## **[Audio starts]**
<audio>
## **[Audio ends]**

Figure 5: Prompt structure for Gemini-2.5-Pro to annotate a single-audio critic.

864
865
866
867
868
869
870
871
872
873
874
875
876
877
878
879
880
881
882
883
884
885
886
887
888
889
890
891
892
893
894
895
896
897
898
899
900
901
902
903
904
905
906
907
908

---

**Prompt for Gemini-2.5-Pro to annotate a paired-audio critic with scores**

**System:**
You are a helpful assistant.

**Prompt:**
## **[Instruction starts]**

You will be given two audio clips.
Please act as a professional speech quality evaluation expert and provide the scoring
rationale based on the following four dimensions:
1. Noise: Whether there is background noise, and whether it interferes with understanding.
2. Distortion: Whether there are compression artifacts, electrical noise, or other distortions.
3. Naturalness: Whether the speech sounds natural and resembles real human speech.
4. Continuity: Whether the speech is fluent and continuous, or if there are any dropouts or
interruptions.

After providing the above scoring rationale, give each audio an overall quality score from
1 to 10 (integer only, 10 = best quality).
The score should be consistent with the justification above.

Finally, output the scores in the exact format:
Final Scores: [[audio1_score, audio2_score]]

## **[Instruction ends]**

## **[Audio1 starts]**
<audio>
## **[Audio1 ends]**

## **[Audio2 starts]**
<audio>
## **[Audio2 ends]**

---

Figure 6: Prompt structure for Gemini-2.5-Pro to annotate a paired-audio critic with scores.

909
910
911
912
913
914
915
916
917

---

**Prompt for scalar and semi-scalar reward models inference**

**System:**
You are a helpful assistant.

**Prompt:**
## **[Instruction starts]**

You will be given an audio clip.
Please act as a professional speech quality evaluation expert and provide an objective description of the input audio based on the following four dimensions:
1. Noise: Whether there is background noise, and whether it interferes with understanding.
2. Distortion: Whether there are compression artifacts, electrical noise, or other distortions.
3. Naturalness: Whether the speech sounds natural and resembles real human speech.
4. Continuity: Whether the speech is fluent and continuous, or if there are any dropouts or interruptions.

## **[Instruction ends]**

## **[Audio starts]**
<audio>
## **[Audio ends]**

---

Figure 7: Prompt structure for scalar and semi-scalar reward models inference.

**Prompt for generative reward models inference**

**System:**
You are a helpful assistant.

**Prompt:**
## **[Instruction starts]**

You will be given two audio clips.
Please act as a professional speech quality evaluation expert and provide the scoring rationale based on the following four dimensions:
1. Noise: Whether there is background noise, and whether it interferes with understanding.
2. Distortion: Whether there are compression artifacts, electrical noise, or other distortions.
3. Naturalness: Whether the speech sounds natural and resembles real human speech.
4. Continuity: Whether the speech is fluent and continuous, or if there are any dropouts or interruptions.

After providing the above scoring rationale, give each audio an overall quality score from 1 to 10 (integer only, 10 = best quality).The score should be consistent with the justification above.

Finally, output the scores in the exact format:
Final Scores: [[audio1_score, audio2_score]]

## **[Instruction ends]**

## **[Audio1 starts]**
<audio>
## **[Audio1 ends]**

## **[Audio2 starts]**
<audio>
## **[Audio2 ends]**

Figure 8: Prompt structure for generative reward models inference.

---

**Prompt for LLM-as-a-judge**

**System:**
You are a helpful assistant.

**Prompt:**
## **[Instruction starts]**

You will be given two audio clips.
Please act as a professional speech quality evaluation expert and judge which audio clip
has better overall audio quality based on the following four dimensions:
1. Noise: Whether there is background noise, and whether it interferes with understanding.
2. Distortion: Whether there are compression artifacts, electrical noise, or other distortions.
3. Naturalness: Whether the speech sounds natural and resembles real human speech.
4. Continuity: Whether the speech is fluent and continuous, or if there are any dropouts or
interruptions.

Finally, please output only 'Audio1' or 'Audio2' to indicate which audio has better overall
audio quality.

## **[Instruction ends]**

## **[Audio1 starts]**
<audio>
## **[Audio1 ends]**

## **[Audio2 starts]**
<audio>
## **[Audio2 ends]**

---

Figure 9: Prompt structure for LLM-as-a-judge.

