# OpenReview forum: "From Scores to Preferences: Redefining MOS Benchmarking for Speech Quality Reward Modeling"
_ICLR.cc/2026/Conference — ICLR 2026 Conference Withdrawn Submission_

### Official Review · Reviewer_ZoDT · 2025-10-16

**Soundness:** 2
**Presentation:** 3
**Contribution:** 2
**Rating:** 2
**Confidence:** 4

**Summary:**

This paper introduces a benchmark called MOS-RMBench, which consists of five in-domain datasets and one out-of-domain dataset. The MOS scores are converted into paired preferences. The paper compares three training strategies on the benchmark and proposes a MOS-aware GRM to enhance the model’s ability to capture fine-grained perceptual quality differences.

**Strengths:**

This paper focuses on an important task: automatically assessing speech quality. This is a crucial research direction for enabling reproducible, consistent, and efficient evaluation of synthesised speech.

**Weaknesses:**

Regarding the benchmark:

1. The authors state that MOS scores “suffer from inconsistent rating standards and poor reproducibility,” but the benchmark construction still relies on MOS scores to derive preference pairs. This raises a conceptual inconsistency.

2. Preference pairs are only formed within each dataset. While I understand that cross-dataset pairing poses difficulties, allowing comparisons across datasets would better leverage the larger data volume and diversity.

3. It seems that no margin filtering was applied when converting MOS differences into binary preferences. Given the noisy nature of MOS annotations, for pairs with very small ΔMOS, it is questionable whether such differences reflect a meaningful perceptual preference even for human listeners. Moreover, the current formulation treats all pairs in the same way—whether the ΔMOS is 0.2 or 2.0—forcing both into a binary positive/negative decision and ignoring the different levels of confidence implied by the margin size. Especially in the situation where, as shown in Figure 2c, a large proportion of pairs lie in the small-margin region (e.g., ΔMOS < 0.5), meaning that ambiguous cases are not the minority but a dominant part of the benchmark. Introducing a threshold to treat small-gap pairs as similar quality might lead to a more meaningful and stable evaluation protocol.


Regarding evaluated models:

4. It seems that only a single backbone model (Qwen2-Audio) was used, which may not be sufficient to support the general conclusions drawn in Section 5.1. It would be more convincing to evaluate both LLM-based and non-LLM-based backbones to verify whether the observed trends and patterns generalise across architectures. In addition, important training details are missing, such as model size and whether the backbone was fully fine-tuned or adapted via PEFT.

5. The terminology “reward model” in Section 4.2 is somewhat confusing. Both classic scalar reward models and semi-scalar reward models appear to refer to supervised fine-tuning strategies applied to the Qwen2-Audio backbone, whereas generative reward models involve RL training with an accuracy-based reward model—effectively introducing a reward model inside another reward model setup. Clarifying this terminology hierarchy would improve conceptual clarity.

6. For GRM, it is stated that “each pair is compared across the same four dimensions and provided with overall quality scores” while the RL reward is binary. It is unclear whether there is any explicit supervision or guidance on how the model should assess the four dimensions or produce the overall score during RL training. Design RL training based on the text description rather than a rule-based reward function might help.

Regarding experiments and results:

7. In Table 2, what is “classic + xx loss” and “cloud + xx loss” means. Explanation seems missing.

8. The authors state that “all models are particularly prone to errors on sample pairs with small MOS differences,” which seems expected given the issue noted earlier: very small ΔMOS values may not correspond to meaningful perceptual preferences and are likely to be dominated by label noise. The observation that “fine-grained discrimination of speech quality remains a critical bottleneck” also appears to follow directly from the decision to polarize all preference pairs regardless of the actual MOS difference.

9. Regarding the MOS-aware reward function, it is not clearly stated where the MOS gap used in the reward function comes from. If the gap is predicted by the model itself, it is unclear how this prediction is trained and, if it were accurate, whether additional RL optimisation would still be necessary. If the gap is instead provided as metadata derived from ground-truth MOS, it raises concerns about practical applicability, as this information would not be available at inference time in real-world usage scenarios.

**Questions:**

See weakness section.

Line 212 "spanning six languages: English, Chinese, Taiwanese Mandarin, Japanese, French, and German." -- not sure whether Chinese and Taiwanese Mandarin should be treated as separate languages or different accents.

---

> ### Author Response · Authors · 2025-11-21
> **Response to Reviewer ZoDT(1/2)**
>
> Thank you for your time and effort in reviewing our work. Based on your comments, we have identified the key issues and provide detailed responses below.
>
> > Weakness1: The authors state that MOS scores "suffer from inconsistent rating standards and poor reproducibility", but the benchmark construction still relies on MOS scores to derive preference pairs. This raises a conceptual inconsistency.
> >
> > Weakness2: Preference pairs are only formed within each dataset. While I understand that cross-dataset pairing poses difficulties, allowing comparisons across datasets would better leverage the larger data volume and diversity.
>
> **Answer:** Thank you for these thoughtful comments. The noted MOS "inconsistency" and "poor reproducibility" refers to differences across datasets, where varying annotation protocols can cause samples of similar quality to receive different MOS scores. This makes cross-dataset MOS values non-comparable. Importantly, this does not imply unreliability within a dataset: each MOS dataset is internally consistent and collected under controlled subjective testing, so there is no conceptual inconsistency.
>
> MOS-RMBench converts within-dataset MOS into pairwise preferences, avoiding cross-dataset mismatches while preserving reliable signals. For the same reason, preference pairs are restricted to within each dataset; cross-dataset pairing would mix incompatible rating scales and introduce bias, producing preference relations that may not reflect true human judgments. We hope this clarification addresses the concern.
>
> > Weakness3: No margin filtering is applied when converting MOS differences into binary preferences. Given the inherent noise in MOS annotations, very small ΔMOS may not reflect reliable perceptual differences. Treating all pairs equally—whether ΔMOS is 0.2 or 2.0—forces a binary choice and ignores confidence differences. As shown in Figure 2c, small-margin pairs (ΔMOS<0.5) constitute a large portion of the data, meaning ambiguous cases dominate. Introducing a threshold to treat small-gap pairs as similar quality could lead to a more meaningful and stable evaluation protocol.
> >
> > Weakness8: The authors state that "all models are particularly prone to errors on sample pairs with small MOS differences", which seems expected given the issue noted earlier: very small ΔMOS values may not correspond to meaningful perceptual preferences and are likely to be dominated by label noise. The observation that "fine-grained discrimination of speech quality remains a critical bottleneck" also appears to follow directly from the decision to polarize all preference pairs regardless of the actual MOS difference.
>
> **Answer:** Thank you for raising these insightful points. Although MOS scores are absolute, they implicitly encode relative perceptual rankings within each dataset. Converting them into preference pairs preserves the underlying human preference structure. Moreover, treating all ΔMOS values as binary outcomes reflects perceptual difficulty: small ΔMOS pairs capture fine-grained distinctions, while large ΔMOS pairs reflect clearer contrasts.
>
> Filtering out small-ΔMOS pairs would bias the benchmark toward easier comparisons and remove the challenging cases that are common in real MOS data, thereby reducing its ability to evaluate models under realistic perceptual conditions.
>
> As an additional analysis, we evaluated two classic reward models on the test set with ΔMOS>0.5: one trained on the full data and one trained on data with ΔMOS>0.5, to examine whether small-ΔMOS pairs affect model performance or evaluation stability. We will report detailed results once the study is complete.
>
> > Weakness4: Relying on a single backbone (Qwen2-Audio) limits the validity of conclusions in Section 5.1; evaluating additional LLM and non-LLM backbones would improve generality. Key training details are missing, such as model size and whether the backbone was fully fine-tuned or adapted via PEFT.
>
> **Answer:** Thank you for raising this concern. While evaluating multiple backbones could strengthen generality, our current choice of Qwen2-Audio as the backbone was motivated by establishing a controlled setting that allows us to isolate the effects of reward modeling techniques rather than architectural differences. Qwen2-Audio also provides strong performance and stable training dynamics, making it a suitable backbone for this initial study.
>
> Regarding the missing training details:
>
> - Backbone size: We use the publicly released Qwen2-Audio-7B version.
> - Training type: All models are trained via full-parameter fine-tuning.
>
> We will revise Section 4.2 (Evaluated Models) and Appendix materials to explicitly include these details.

---

> > ### Author Response · Authors · 2025-11-21
> > **Response to Reviewer ZoDT(2/2)**
> >
> > > Weakness5: The terminology "reward model" in Section 4.2 is confusing. Both classic scalar reward models and semi-scalar reward models appear to refer to supervised fine-tuning strategies applied to the Qwen2-Audio backbone, whereas generative reward models involve RL training with an accuracy-based reward model—effectively introducing a reward model inside another reward model setup. Clarifying this terminology hierarchy would improve conceptual clarity.
> >
> > **Answer:** Thank you for pointing out this issue and for the helpful suggestion. We would like to clarify that all three paradigms (Classic scalar, Semi-scalar, and GRMs) use the same Qwen2-Audio backbone. During the RL stage, GRMs are optimized with an accuracy-based reward (e.g., GRPO or DAPO) rather than a separate reward model—i.e., there is no "reward model inside another reward model".
> >
> > We will revise Section 4.2 (Evaluated Models) in the updated manuscript to clarify the evaluated models. We hope this addresses the concern.
> >
> > > Weakness6: For GRM, it is stated that "each pair is compared across the same four dimensions and provided with overall quality scores" while the RL reward is binary. It is unclear whether there is any explicit supervision or guidance on how the model should assess the four dimensions or produce the overall score during RL training. Design RL training based on the text description rather than a rule-based reward function might help.
> >
> > **Answer:** Thank you for the valuable comment. During the SFT stage, the GRM is explicitly supervised to generate critiques along the four perceptual dimensions (noise, distortion, naturalness, continuity) and to output overall quality scores for each sample. Thus, the model learns how to assess these dimensions before RL. The RL stage then aligns the model with human preferences using a rule-based reward derived from the predicted overall scores, following DeepSeek-GRM [1]. This reward is fully consistent with MOS-RMBench’s pairwise accuracy metric and provides a stable, dataset-agnostic training objective.
> >
> > While designing an RL signal directly from textual descriptions is an interesting direction, textual critiques are inherently free-form and can vary in length and phrasing, which make it difficult to construct a reliable, low-noise reward. In contrast, the rule-based reward grounded in MOS-derived preference labels is more robust and interpretable in our setting.
> >
> > We sincerely thank you for your suggestion and hope this clarification helps address the concern.
> >
> > > Weakness7: In Table 2, what is "classic + xx loss" and "cloud + xx loss" means. Explanation seems missing.
> >
> > **Answer:** Thank you for pointing this out. We apologize for the missing clarification.
> >
> > - "Classic + xx loss" refers to training the classic scalar reward models with either the BT loss or the MSE loss: the BT loss encourages the chosen sample to score higher than the rejected one; the MSE loss regresses the predicted score to the MOS value.
> > - "Cloud + xx loss" refers to training the semi-scalar reward models with the same two losses (BT or MSE).
> >
> > We will revise Table 2 and Section 4.2 (Evaluated Models) to explicitly define these terms and describe the corresponding loss functions.
> >
> > > Weakness9: It is unclear where the MOS gap in the reward function comes from. If predicted by the model, its training process and the need for further RL optimization are unclear. If derived from ground-truth MOS, it may not be practical, as such information is unavailable at inference.
> >
> > **Answer:** Thank you for raising this question. We apologize for the lack of clarity. The MOS gap in the MOS-aware reward is not predicted by the model and is not required at inference. It is used only during RL training, computed from the ground-truth MOS difference within each training pair. The GRM never receives MOS as input and does not rely on it at test time, ensuring practical applicability.
> >
> > We will revise Section 5.3 (MOS-aware GRM) to clarify the source and scope of the MOS-aware reward.
> >
> > > Question1: Line 212 "spanning six languages: English, Chinese, Taiwanese Mandarin, Japanese, French, and German." -- not sure whether Chinese and Taiwanese Mandarin should be treated as separate languages or different accents.
> >
> > **Answer:** Thank you for pointing out this issue. In our manuscript, "Chinese" refers to Mandarin, while "Taiwanese Mandarin" refers to the Mandarin variety spoken in Taiwan. We fully agree that these two should not be treated as different languages.
> >
> > To avoid any misunderstanding, we will revise Section 3.3 (Dataset Statistics) to state that MOS-RMBench covers five languages: English, Chinese, Japanese, French, and German, and we will update Table 1 accordingly by changing the TMHINT-QI language label to Chinese.
> >
> > We hope this resolves the concern.
> >
> > ------
> >
> > **Reference**
> >
> > [1] Liu Z, Wang P, Xu R, et al. Inference-time scaling for generalist reward modeling[J]. arXiv preprint arXiv:2504.02495, 2025.

---

> ### Author Response · Authors · 2025-11-26
> **Response to Reviewer ZoDT (Results and Analysis of the Supplementary Experiments)**
>
> > Weakness3: No margin filtering is applied when converting MOS differences into binary preferences. Given the inherent noise in MOS annotations, very small ΔMOS may not reflect reliable perceptual differences. Treating all pairs equally—whether ΔMOS is 0.2 or 2.0—forces a binary choice and ignores confidence differences. As shown in Figure 2c, small-margin pairs (ΔMOS<0.5) constitute a large portion of the data, meaning ambiguous cases dominate. Introducing a threshold to treat small-gap pairs as similar quality could lead to a more meaningful and stable evaluation protocol.
> >
> > Weakness8: The authors state that "all models are particularly prone to errors on sample pairs with small MOS differences", which seems expected given the issue noted earlier: very small ΔMOS values may not correspond to meaningful perceptual preferences and are likely to be dominated by label noise. The observation that "fine-grained discrimination of speech quality remains a critical bottleneck" also appears to follow directly from the decision to polarize all preference pairs regardless of the actual MOS difference.
>
> **Answer:** Thank you for raising these insightful points. To directly examine whether small-ΔMOS pairs negatively impact model performance or evaluation stability, we conducted additional experiments by removing all pairs with ΔMOS ≤ 0.5 from the test set and training two Classic reward models with BT Loss:
>
> (1) one trained on the full training data, and
>
> (2) one trained on a filtered training set containing only pairs with ΔMOS > 0.5.
>
> Both models were evaluated on the ΔMOS > 0.5 test set, and the evaluation results are presented as follows:
>
> | Model                         |   BVCC    |   NISQA   |  SingMOS  |   SOMOS   | TMHINT-QI |  VMC'23   |  Overall  |
> | :---------------------------- | :-------: | :-------: | :-------: | :-------: | :-------: | :-------: | :-------: |
> | Classic+BT Loss(w/o ∆MOS≤0.5) |   92.27   |   90.30   | **81.06** |   86.04   | **87.73** |   84.43   |   87.00   |
> | Classic+BT Loss(full data)    | **93.13** | **91.06** |   80.92   | **87.38** |   87.20   | **86.27** | **87.90** |
>
> The results show that the model trained on the full data consistently outperforms the model trained without small-ΔMOS pairs across most datasets and in overall accuracy. This indicates that small-margin pairs do not introduce harmful noise or degrade model performance; instead, they help the model learn finer-grained perceptual distinctions.
>
> Therefore, small-ΔMOS pairs are an inherent characteristic of human MOS distributions, are not dominated by noise, and do not destabilize evaluation. Retaining them preserves the realistic perceptual difficulty of the task and enables the benchmark to assess fine-grained quality discrimination.
>
> We hope these additional results address the concern and we would be delighted to receive any additional suggestions or comments.

---

### Official Review · Reviewer_3TuS · 2025-10-31

**Soundness:** 3
**Presentation:** 3
**Contribution:** 3
**Rating:** 4
**Confidence:** 3

**Summary:**

This paper addresses the limitations of traditional Mean Opinion Score (MOS) evaluations—such as inconsistent standards and poor reproducibility—by introducing MOS-RMBench, a unified benchmark that transforms heterogeneous MOS datasets into a preference-comparison framework. The authors systematically evaluate three reward modeling paradigms (scalar, semi-scalar, generative) and identify key challenges: scalar models achieve the highest overall accuracy (~80%), most models underperform on synthetic speech, and all struggle with fine-grained discrimination for pairs with small MOS differences. To mitigate this, a MOS-aware Generative Reward Model (GRM) is proposed, which incorporates MOS-difference-based rewards to adaptively scale learning signals. Experimental results show consistent performance improvements, especially on challenging pairs, establishing both a benchmark and methodological foundation for speech quality assessment.

**Strengths:**

Novel Unified Benchmark: MOS-RMBench fills a critical gap by integrating six diverse MOS datasets into a consistent preference-based framework, enabling rigorous cross-dataset and cross-domain evaluation—an advancement over fragmented existing resources.
Comprehensive Paradigm Comparison: The systematic evaluation of scalar, semi-scalar, and generative reward models, along with LLM-as-a-judge and MOS prediction baselines, provides valuable insights into their relative strengths and weaknesses for speech quality assessment.
Targeted Innovation with MOS-aware GRM: The proposed MOS-difference-based reward function directly addresses the core challenge of fine-grained quality discrimination, delivering measurable improvements on high-difficulty sample pairs and enhancing model interpretability.
Rigorous Methodology: The paper demonstrates strong reproducibility through detailed dataset statistics, training configurations, prompt templates, and error analysis, adhering to high academic standards for empirical research.

**Weaknesses:**

Limited Language and Quality Dimensions: MOS-RMBench covers only six languages and focuses solely on perceptual quality, neglecting important dimensions like prosody, emotion, and style—key aspects of real-world speech generation evaluation.
Potential Bias from LLM Annotations: The reliance on Gemini-2.5-Pro for generating natural-language critiques and preference labels introduces potential annotation biases, which are not fully discussed or validated against human judgments.
Narrow Analysis of Domain Gap: While the paper notes poorer performance on synthetic speech, it lacks in-depth exploration of the root causes (e.g., acoustic differences, data distribution shifts) or strategies to bridge this gap beyond the proposed GRM.

**Questions:**

How does the annotation quality of Gemini-2.5-Pro compare to human annotators for preference labeling and critique generation? Are there any cases where LLM annotations diverge significantly from human judgments, and how might this impact model training?
Given the focus on fine-grained discrimination, have the authors tested the MOS-aware GRM on pairs with extremely small MOS differences (e.g., ∆MOS < 0.2) to further validate its effectiveness in the most challenging scenarios?

---

> ### Author Response · Authors · 2025-11-21
> **Response to Reviewer 3TuS(1/2)**
>
> Thank you for your time and effort in reviewing our work. Based on your comments, we have identified the key issues and provide detailed responses below.
>
> > Weakness1: Limited Language and Quality Dimensions: MOS-RMBench covers only six languages and focuses solely on perceptual quality, neglecting important dimensions like prosody, emotion, and style—key aspects of real-world speech generation evaluation.
>
> **Answer:** Thank you for the thoughtful comment. While prosody, emotion, and style are important in speech generation, MOS-RMBench is deliberately scoped to speech quality as measured by MOS (e.g., naturalness, clarity, fluency). This is because (1) MOS is the most widely adopted subjective metric in speech domain, and (2) large-scale MOS datasets with consistent annotation procedures enable a scalable, reproducible benchmark. Prosody, emotion, and style use annotation protocols that differ from MOS, making unification challenging.
>
> Regarding language coverage, MOS-RMBench includes five languages (treating Taiwanese Mandarin as a dialect), which is already broader than prior MOS-based benchmarks. Expanding to more languages is primarily limited by data availability rather than framework design. We hope this clarifies the concern.
>
> > Weakness2: Potential Bias from LLM Annotations: The reliance on Gemini-2.5-Pro for generating natural-language critiques and preference labels introduces potential annotation biases, which are not fully discussed or validated against human judgments.
> >
> > Question1: How does the annotation quality of Gemini-2.5-Pro compare to human annotators for preference labeling and critique generation? Are there any cases where LLM annotations diverge significantly from human judgments, and how might this impact model training?
>
> **Answer:** Thank you for raising this important point. The critiques and quality scores (align with ground-truth preference ordering) generated by Gemini-2.5-Pro are used only during the SFT stage of semi-scalar reward models and GRMs, and do not contribute to the creation of preference labels. All preference labels are derived from ground-truth MOS scores, which serve as the primary supervision for both scalar/semi-scalar model training and the RL stage of GRMs. Therefore, LLM annotations do not affect the core preference supervision or the evaluation. We also mitigate potential LLM bias through deterministic prompting, constraining critique dimensions, and randomly swapping items within each preference pair.
>
> To further assess reliability, we are conducting human validation on 100 randomly sampled items, with four evaluators judging whether Gemini-2.5-Pro’s critiques align with their own perceptions. Detailed results and any divergences from human judgments will be reported once the study is complete.
>
> > Question2: Given the focus on fine-grained discrimination, have the authors tested the MOS-aware GRM on pairs with extremely small MOS differences (e.g., ∆MOS < 0.2) to further validate its effectiveness in the most challenging scenarios?
>
> **Answer:** Thank you for the question. We are currently conducting additional experiments focusing specifically on this subset. These experiments evaluate the MOS-aware GRM’s accuracy on small-margin pairs to further validate its effectiveness. We will provide the detailed results and accompanying analysis as soon as they are available.

---

> > ### Author Response · Authors · 2025-11-21
> > **Response to Reviewer 3TuS(2/2)**
> >
> > > Weakness3: Narrow Analysis of Domain Gap: While the paper notes poorer performance on synthetic speech, it lacks in-depth exploration of the root causes (e.g., acoustic differences, data distribution shifts) or strategies to bridge this gap beyond the proposed GRM.
> >
> > **Answer:** Thank you for the insightful comment. We attribute this gap to two main factors:
> >
> > (1) Acoustic differences. Human-speech-based datasets (e.g., TMHINT-QI, NISQA) primarily consist of real human recordings or controlled degradations, which are straightforward and perceptually salient, making them relatively easier for reward models to learn. In contrast, synthetic-speech-based datasets often contain more subtle or complex artifacts, which are harder to model accurately.
> >
> > (2) Data distribution differences. We statistics the ∆MOS distributions across different datasets:
> >
> > |  Dataset  | <=0.5 | 0.5~1.0 | 1.0~1.5 | 1.5~2.0 | 2.0~2.5 | 2.5~3.0 | 3.0~3.5 | 3.5~4.0 |
> > | :-------: | :---: | :-----: | :-----: | :-----: | :-----: | :-----: | :-----: | :-----: |
> > |   BVCC    | 0.30  |  0.24   |  0.21   |  0.13   |  0.08   |  0.03   |  0.01   |  0.00   |
> > |   NISQA   | 0.23  |  0.20   |  0.19   |  0.14   |  0.10   |  0.06   |  0.04   |  0.03   |
> > |  SingMOS  | 0.28  |  0.38   |  0.17   |  0.14   |  0.02   |  0.02   |  0.00   |  0.00   |
> > |   SOMOS   | 0.48  |  0.34   |  0.14   |  0.03   |  0.01   |  0.00   |  0.00   |  0.00   |
> > | TMHINT-QI | 0.24  |  0.30   |  0.15   |  0.18   |  0.06   |  0.05   |  0.02   |  0.00   |
> > |  VMC'23   | 0.33  |  0.25   |  0.18   |  0.12   |  0.07   |  0.04   |  0.01   |  0.00   |
> >
> > Synthetic-speech-based datasets show a significantly higher proportion of samples in the most challenging region (∆MOS ≤ 0.5), 0.48 for SOMOS and 0.33 for VMC'23, compared to human-speech-based datasets, which have 0.24 for TMHINT-QI and 0.23 for NISQA. This confirms that synthetic-speech datasets are more challenging, as the differences in quality between samples are finer and more difficult to discriminate.
> >
> > Notably, the primary goal of our work is to introduce a unified benchmark and to evaluate existing reward-modeling paradigms, rather than to propose new modeling approaches. Nevertheless, these observations highlight challenges that could guide future research on bridging the domain gap.

---

> ### Author Response · Authors · 2025-11-26
> **Response to Reviewer 3TuS (Results and Analysis of the Supplementary Experiments)**
>
> > Weakness2: Potential Bias from LLM Annotations: The reliance on Gemini-2.5-Pro for generating natural-language critiques and preference labels introduces potential annotation biases, which are not fully discussed or validated against human judgments.
> >
> > Question1: How does the annotation quality of Gemini-2.5-Pro compare to human annotators for preference labeling and critique generation? Are there any cases where LLM annotations diverge significantly from human judgments, and how might this impact model training?
>
> **Answer:** Thank you for raising this important question. We have now completed the human validation experiment referenced earlier. Four human evaluators assessed whether Gemini-2.5-Pro’s critiques were consistent with their own perceptual judgments on 100 randomly sampled items. We observed that 70% of the critiques were judged as aligned with human perception.
>
> Unlike binary preference judgments, natural-language critiques require describing fine-grained perceptual attributes (e.g., noise, distortion, naturalness, continuity), where subjective variability is inevitable. Under such conditions, a 70% human–LLM alignment rate is well within a reasonable and expected range.
>
> Overall, the LLM-generated critiques are broadly consistent with human perceptual impressions. More importantly, LLM annotations do not affect the core preference supervision or the evaluation.
>
> > Question2: Given the focus on fine-grained discrimination, have the authors tested the MOS-aware GRM on pairs with extremely small MOS differences (e.g., ∆MOS < 0.2) to further validate its effectiveness in the most challenging scenarios?
>
> **Answer:** Thank you for the question. To directly evaluate whether the MOS-aware GRM improves fine-grained discrimination, we conducted additional experiments focusing specifically on the most challenging subset: sample pairs with extremely small MOS gaps (∆MOS < 0.2). We tested both GRPO-trained and DAPO-trained GRMs and compared the MOS-aware GRMs against the vanilla GRMs.
>
> The evaluation results are presented as follows:
>
> **GRPO setting:**
>
> | Model              |   BVCC    |   NISQA   |  SingMOS  |   SOMOS   | TMHINT-QI |  VMC'23   |  Overall  |
> | :----------------- | :-------: | :-------: | :-------: | :-------: | :-------: | :-------: | :-------: |
> | GRM+GRPO           |   56.79   |   52.85   |   60.98   | **58.19** |   43.24   |   53.20   |   54.41   |
> | MOS-aware GRM+GRPO | **58.02** | **54.40** | **63.41** |   57.63   | **48.65** | **53.78** | **55.33** |
>
> **DAPO setting:**
>
> | Model              | BVCC  |   NISQA   | SingMOS |   SOMOS   | TMHINT-QI |  VMC'23   |  Overall  |
> | :----------------- | :---: | :-------: | :-----: | :-------: | :-------: | :-------: | :-------: |
> | GRM+DAPO           | 58.02 |   54.40   |  63.41  |   57.06   |   48.65   |   52.62   |   54.75   |
> | MOS-aware GRM+DAPO | 58.02 | **55.44** |  63.41  | **59.32** |   48.65   | **53.49** | **55.78** |
>
> Across both training algorithms, the MOS-aware GRMs show consistently improved or comparable accuracy on pairs with extremely small MOS differences:
>
> - GRPO setting: Compared with the vanilla GRM, the MOS-aware GRM improves overall accuracy from 54.41 to 55.33, with gains on BVCC, NISQA, SingMOS, TMHINT-QI, and VMC’23. A slight decrease is observed on SOMOS, but the gap is negligible.
> - DAPO setting: The MOS-aware GRM improves overall accuracy from 54.75 to 55.78, showing equal or better performance across all datasets.
>
> These results demonstrate that incorporating MOS-gap information into the reward function yields measurable benefits specifically in the fine-grained regime where perceptual differences are extremely small and the task is most challenging.
>
> We hope these additional results effectively address the concern, and we would be delighted to receive any additional suggestions or comments.

---

### Official Review · Reviewer_MjVg · 2025-11-01

**Soundness:** 2
**Presentation:** 3
**Contribution:** 2
**Rating:** 4
**Confidence:** 3

**Summary:**

The work is well-motivated: MOS is costly, inconsistent across datasets, and difficult to unify; preference-based framing is more comparable. The benchmark integrates six datasets spanning multiple languages, vocoder types, and tasks (TTS, SE, VC, SVC, singing, noisy/reverberant speech). It introduces MOS-RMBench, a benchmark that reformulates multiple MOS datasets into pairwise preference comparisons to enable consistent training and evaluation of speech-quality reward models. Based on this unified preference view, the authors benchmark three reward-modeling paradigms — scalar, semi-scalar, and generative reward models (GRMs) — implemented on Qwen2-Audio. Major findings include: (i) scalar RMs perform best overall, reaching ~80% accuracy; (ii) models perform substantially worse on synthetic speech; and (iii) all models struggle when MOS differences are small. To address this, the authors propose a MOS-aware GRM that incorporates a MOS-difference-dependent reward shaping term, improving discrimination on difficult pairs.

**Strengths:**

* A very interesting study tackling a difficult problem, with a commendable effort to collate and repurpose existing MOS crowdsourced annotation (which are generally low-resource and heterogeneous) into a more structured preference-based benchmark.
* Systematic evaluation of three RM paradigms; the proposed formulation of classic, semi-scalar, and generative reward modeling is well-structured and novel as a comparative framework.

**Weaknesses:**

* **(No New Human Preference Labels)** All labels originate from MOS and are reinterpreted into preferences. Without new preference annotation, MOS-derived ordering is assumed valid at instance level, even though MOS reliability is often debated at utterance granularity. This risks embedding existing scoring noise and dataset-specific biases directly into the benchmark.
* **(Pair Construction Rules May Create Artificial Comparisons)** The benchmark groups samples not only by shared content but also by shared speaker or system. Grouping by content aligns with CMOS/MUSHRA practices, but grouping by speaker or system is less natural for perceptual quality comparison, since two samples may differ semantically and still be forced into a preference pair. It is unclear that such comparisons reflect meaningful perceptual judgments. Moreover, MOS is a score reported after averaging across multiple utterances and still widely criticized; here, using utterance-level MOS to infer pairwise preferences introduces additional unreliability.
* **(No External Validation of Benchmark Quality)** There is no evidence of human validation of the resulting preference pairs or benchmark sanity checking. Without external human verification or comparison against established tests (e.g., MUSHRA/CCR), it is difficult to assess whether these preference labels are perceptually valid. One indicator would be correlation of benchmark-derived rankings with other evaluation protocols.
* **(No Study of Preference Reversals or Ranking Fidelity)** Accuracy is reported, but not whether predicted preferences preserve global ranking or avoid cycles

**Questions:**

1. How are “unreliable annotations” identified and filtered? The paper states that samples are filtered but does not describe the criteria, annotation confidence, or thresholds used.

2. Pair construction: grouping by shared content makes sense, but grouping by shared speaker/system seems less principled. What is the perceptual motivation? Why is this considered a meaningful comparison when MOS was not collected in a paired setting?

3. Was any human review or quality check conducted on a subset of constructed preference pairs to evaluate benchmark validity?

4. Is there evidence that model rankings on MOS-RMBench correlate with rankings from other benchmarks?

---

> ### Author Response · Authors · 2025-11-21
> **Response to Reviewer MjVg(1/2)**
>
> Thank you for your time and effort in reviewing our work. Based on your comments, we have identified the key issues and provide detailed responses below.
>
> > Weakness1: **(No New Human Preference Labels)** All labels originate from MOS without new preference annotations. Assuming MOS ordering is valid at the instance level may embed scoring noise and dataset-specific biases into the benchmark.
> >
> > Weakness2: Moreover, MOS is a score reported after averaging across multiple utterances and still widely criticized; here, using utterance-level MOS to infer pairwise preferences introduces additional unreliability.
>
> **Answer:** Thank you for the thoughtful comment. We acknowledge that MOS may contain utterance-level noise, but MOS-RMBench is explicitly designed to mitigate this. By using pairwise preference comparisons instead of absolute MOS values, our approach reduces the impact of annotator bias. All preference pairs are constructed within each dataset, preserving internal scoring standards and avoiding dataset-specific scale effects.
>
> Our goal is to build a scalable, reproducible benchmark across multiple datasets and speech domains. Collecting new human preference labels at this scale would require hundreds of hours of expert listening, which is impractical. Reinterpreting existing MOS labels as preference signals allows us to achieve this goal while maintaining consistency and broad applicability. The benchmark is also fully compatible with future datasets containing human preference judgments.
>
> > Weakness2: **(Pair Construction Rules May Create Artificial Comparisons)** The benchmark groups samples not only by shared content but also by shared speaker or system. Grouping by content aligns with CMOS/MUSHRA practices, but grouping by speaker or system is less natural for perceptual quality comparison, since two samples may differ semantically and still be forced into a preference pair. It is unclear that such comparisons reflect meaningful perceptual judgments.
> >
> > Question2: Pair construction: grouping by shared content makes sense, but grouping by shared speaker/system seems less principled. What is the perceptual motivation?
>
> **Answer:** Thank you for raising this concern. Our grouping strategy is based on the characteristics of each dataset. For most datasets we use, samples share the same content while differing in speaker/system. However, a few datasets feature identical speaker/system conditions but varying content (such as NISQA, which contains real telephone recordings from the same speaker).
>
> Previous studies have shown that even under identical speaker/system conditions, human listeners still perceive quality differences across samples (e.g., NISQA; VoiceMOS Challenge 2023 [1]). We view this perceptual difference as reflecting genuine human preference patterns. Moreover, real-world speech-quality assessment is not limited to matched-content scenarios.
>
> Given the structure of existing MOS datasets, our grouping strategy avoids cross-dataset comparisons while preserving meaningful perceptual contrasts, allowing us to build a unified benchmark that better evaluates model behavior in practical settings.
>
> > Question2: Why is this considered a meaningful comparison when MOS was not collected in a paired setting?
>
> **Answer:** Thank you for the question. Although MOS is not collected in pairs, it inherently encodes listeners’ comparative judgments within each dataset: samples with higher MOS are consistently perceived as higher quality than those with lower MOS. Our construction makes this relative structure explicit by converting MOS scores into pairwise preferences, without adding assumptions beyond the MOS protocol. This approach of converting absolute quality scores into relative preference labels is a well-established paradigm in speech preference modeling[2] [3]. We hope this clarification addresses the concern.
>
> ------
>
> **Reference**
>
> [1] Cooper E, Huang W C, Tsao Y, et al. The VoiceMOS Challenge 2023: Zero-shot subjective speech quality prediction for multiple domains[C]//2023 IEEE Automatic Speech Recognition and Understanding Workshop (ASRU). IEEE, 2023: 1-7.
>
> [2] Wang K, Zhao Y, Dong Q, et al. MOSPC: MOS prediction based on pairwise comparison[J]. arXiv preprint arXiv:2306.10493, 2023.
>
> [3] Shi Y F, Ai Y, Ling Z H. Universal Preference-Score-based Pairwise Speech Quality Assessment[J]. arXiv preprint arXiv:2506.01455, 2025.

---

> > ### Author Response · Authors · 2025-11-21
> > **Response to Reviewer MjVg(2/2)**
> >
> > > Weakness3: **(No External** **Validation** **of Benchmark Quality)** There is no evidence of human validation of the resulting preference pairs or benchmark sanity checking. Without external human verification or comparison against established tests (e.g., MUSHRA/CCR), it is difficult to assess whether these preference labels are perceptually valid. One indicator would be correlation of benchmark-derived rankings with other evaluation protocols.
> > >
> > > Question3: Was any human review or quality check conducted on a subset of constructed preference pairs to evaluate benchmark validity?
> > >
> > > Question4: Is there evidence that model rankings on MOS-RMBench correlate with rankings from other benchmarks?
> >
> > **Answer:** We sincerely thank you for the insightful comments and valuable suggestions regarding the external validation of benchmark quality.
> >
> > **Human** **validation****:** All MOS datasets in MOS-RMBench were validated or filtered during their original construction. For example, SOMOS includes human inspection to remove unreliable labels, yielding the SOMOS-clean subset. TMHINT-QI validates ratings by comparing listening-test scores with objective metrics, showing that higher SNR correlates with better perceived quality and intelligibility. MOS-RMBench reorganizes these validated MOS scores into preference pairs, ensuring perceptual validity and alignment with human preferences. To further support this, we are conducting an additional human validation study in which four evaluators label "chosen/rejected" on 100 randomly sampled preference pairs. We will report the correlation between human judgments and benchmark labels once complete.
> >
> > **Correlation with other evaluation protocols:** MUSHRA, CCR, and MOS are all valid methods for assessing perceived speech quality, but MOS offers the broadest coverage and the largest publicly available datasets. To check consistency between model rankings on MOS-RMBench and other protocols, we are constructing preference pairs on the MANGO dataset [1], based on the MUSHRA method, and evaluating the models from our paper. Full results and the correlation between rankings on MANGO and MOS-RMBench will be reported once available.
> >
> > > Weakness4: **(No Study of Preference Reversals or Ranking Fidelity)** Accuracy is reported, but not whether predicted preferences preserve global ranking or avoid cycles
> >
> > **Answer:** Thank you for raising this important point regarding global ranking and preference cycles.
> >
> > Evaluating global ranking requires comparing pairs across different content, speakers, or acoustic conditions, which goes beyond the grouping principles used in MOS-RMBench.
> >
> > Regarding preference cycles, the classic and semi-classic reward models assign deterministic scalar scores independently (temperature = 0), inherently preventing cycles. GRM models score each pair directly and could produce cycles. However, MOS-RMBench is designed as a pairwise evaluation benchmark, and accuracy as the main metric reflects alignment with human pairwise preferences. Any preference cycles would reduce accuracy, serving as a conservative measure of consistency.
> >
> > To provide further insight, we are analyzing potential preference cycles in GRM models by constructing triplets within content-based groups and evaluating three-way cycle rates alongside pairwise accuracy. Full results will be reported once available.
> >
> > > Question1: How are “unreliable annotations” identified and filtered? The paper states that samples are filtered but does not describe the criteria, annotation confidence, or thresholds used.
> >
> > **Answer:** We sincerely thank you for the comment regarding data processing, which allows us to clarify key steps in benchmark construction.
> >
> > Filtering in MOS-RMBench focuses on data integrity and format correctness. We excluded samples with incomplete, missing, or misformatted metadata, which are required for pairwise preference construction (e.g., speaker ID, content ID, system ID).
> >
> > We will revise Section 3.2 (Preference Annotation) in the updated manuscript to explicitly detail these filtering criteria, ensuring clarity on our quality-control process.
> >
> > ------
> >
> > **Reference**
> >
> > [1] Varadhan P S, Gulati A, Sankar A, et al. Rethinking MUSHRA: Addressing Modern Challenges in Text-to-Speech Evaluation[J]. arXiv preprint arXiv:2411.12719, 2024.

---

> ### Author Response · Authors · 2025-11-26
> **Response to Reviewer MjVg (Results and Analysis of the Supplementary Experiments)**
>
> > Weakness3: **(No External** **Validation** **of Benchmark Quality)** There is no evidence of human validation of the resulting preference pairs or benchmark sanity checking. Without external human verification or comparison against established tests (e.g., MUSHRA/CCR), it is difficult to assess whether these preference labels are perceptually valid. One indicator would be correlation of benchmark-derived rankings with other evaluation protocols.
> >
> > Question3: Was any human review or quality check conducted on a subset of constructed preference pairs to evaluate benchmark validity?
> >
> > Question4: Is there evidence that model rankings on MOS-RMBench correlate with rankings from other benchmarks?
>
> **Answer:** We sincerely thank you for the insightful comments.
>
> **Human validation:** We have completed the human validation study referenced earlier. Four human evaluators annotated "chosen/rejected" labels for 100 randomly sampled preference pairs, yielding an 82% agreement with the benchmark’s MOS-derived labels. Given the inherent variability in subjective listening assessments, especially for fine-grained quality differences, this level of agreement indicates that the benchmark closely reflects human perceptual preferences. We will include these results and analyses in the revised manuscript.
>
> **Correlation with other evaluation protocols:** We have completed the correlation experiment referenced earlier. Using the English subset of the MANGO dataset (constructed with the MUSHRA method), we created 120 preference pairs and evaluated all models. The results are summarized as follows:
>
> | Model              | MANGO-English |
> | ------------------ | :-----------: |
> | UTMOS              |     80.83     |
> | Gemini-2.5-Pro     |     65.83     |
> | Qwen2.5-Omni-7B    |     48.33     |
> | Classic + BT Loss  |     80.83     |
> | Classic + MSE Loss |     69.17     |
> | Cloud + BT Loss    |     70.00     |
> | Cloud + MSE Loss   |     60.83     |
> | GRM + SFT          |     66.67     |
> | GRM + GRPO         |     68.33     |
> | GRM + DAPO         |     77.50     |
>
> Across all models, the Spearman correlation between rankings on MANGO-English and MOS-RMBench is 0.515 (p = 0.128), indicating a moderate agreement. Notably, when considering only the three reward-modeling paradigms (the seven models corresponding to Classic scalar, Semi-scalar, and GRM), the correlation increases to 0.714 (p = 0.071), suggesting that the relative ranking of reward models is consistent across datasets. These results provide evidence that the MOS-RMBench preference pairs are perceptually valid and reasonably aligned with the MUSHRA-based dataset.
>
> > Weakness4: **(No Study of Preference Reversals or Ranking Fidelity)** Accuracy is reported, but not whether predicted preferences preserve global ranking or avoid cycles
>
> **Answer:** Thank you for raising this important concern. To directly evaluate whether GRMs exhibit preference cycles, we conducted an additional study using explicit triplet-based cycle detection. For each dataset, we followed the original grouping rules and sampled triplets (A, B, C) with distinct MOS scores such that MOS(A) > MOS(B) > MOS(C), yielding three preference pairs: (A,B), (B,C), and (A,C). In total, we constructed 565 triplets (1695 preference pairs) across all datasets (65 from SOMOS and 100 from each of the others).
>
> We evaluated GRMs trained with SFT, GRPO and DAPO:
>
> **(1) Pairwise Accuracy (%):**
>
> | Model    | BVCC  | NISQA | SingMOS | SOMOS | TMHINT-QI | VMC'23 | Overall |
> | :------- | :---: | :---: | :-----: | :---: | :-------: | :----: | :-----: |
> | GRM+SFT  | 82.67 | 85.00 |  73.67  | 64.62 |   80.00   | 66.33  |  76.05  |
> | GRM+GRPO | 84.67 | 85.33 |  74.67  | 72.31 |   77.33   | 71.33  |  77.94  |
> | GRM+DAPO | 84.00 | 87.00 |  75.67  | 70.77 |   81.00   | 72.33  |  78.94  |
>
> **(2) Preference-cycle Rate (%):**
>
> | Model    | BVCC | NISQA | SingMOS | SOMOS | TMHINT-QI | VMC'23 | Overall |
> | :------- | :--: | :---: | :-----: | :---: | :-------: | :----: | :-----: |
> | GRM+SFT  | 2.00 | 2.00  |  2.00   | 1.54  |   0.00    |  9.00  |  2.83   |
> | GRM+GRPO | 0.00 | 0.00  |  0.00   | 0.00  |   0.00    |  1.00  |  0.18   |
> | GRM+DAPO | 0.00 | 0.00  |  0.00   | 0.00  |   1.00    |  0.00  |  0.18   |
>
> The results show that:
>
> - The SFT-trained GRM exhibits a small but non-negligible cycle rate (2.83% overall) alongside lower pairwise accuracy.
> - Both RL-trained GRMs (GRPO and DAPO) achieve substantially higher accuracy and reduce preference cycles to almost zero (overall 0.18% for both methods), with only a single cycle observed across all datasets.
>
> These results indicate that RL-based training improves alignment with human preferences and substantially reduces preference reversals. We will include these results and analyses in the revised manuscript.
>
> We hope these additional results effectively address the concern, and we would be delighted to receive any additional suggestions or comments.

---

### Note · Authors · 2026-01-04

I have read and agree with the venue's withdrawal policy on behalf of myself and my co-authors.